# Adversarial Robustness based on Randomized Smoothing in Quantum Machine Learning

## Abstract

We present an end-to-end Quantum Machine Learning algorithm for Quantum Adversarial Robustness (QuAdRo) that provides a certified radius for a base classifier, with robustness guarantees based on randomized smoothing - the state-of-the-art defense against adversarial attacks. Classically, the number of samples, also the number of queries to the base classifier scale with $O(1/\epsilon^2)$ where $\epsilon$ is the desired error bound in the expected value of the probability measure $\rho$ defined over the randomized smoothing neighborhood around the input. Our algorithm solves the same problem for a Quantum Computing classifier. We prove that the number of queries to the base classifier is $O(1/\epsilon)$ for the same confidence and error bound. We also present the unitary circuit for QuAdRo, which includes the state preparation methods and circuits for smoothing distributions used to defend against common adversaries - modelled using $l_0, l_1, l_2$ norms, and other metrics. The results of the comparison between the classical and the simulation of the quantum algorithm are also discussed.

## 1 Introduction

Machine Learning (ML) models have become ubiquitous over the last decade. There has also been a massive interest in Quantum Computing (QC) and Quantum Machine Learning (QML) (Aaronson, 2015), with algorithms like Shor's factoring (Shor, 1999), HHL (Harrow et al., 2009) algorithm for solving a linear system of equations, etc providing an exponential speedup over their classical counterparts. There exists another class of QC algorithms with a polynomial speedup - Grover's algorithm (Grover, 1996) for random database search, Quantum Amplitude Estimation (QAE) (Brassard et al., 2002) for counting problem, Bernstein-Vazirani (Bernstein & Vazirani, 1997) algorithm for parity problem, etc.

Shortcomings of classical ML algorithms against malicious actors is a widely-studied sub-domain (Goodfellow et al., 2014; Madry et al., 2017) of ML. Common attack vectors include data poisoning, backdoor attacks, and adversarial attacks. Adversarial attacks can easily trick well-trained classifiers into misclassifying an input perturbed by a small, usually imperceptible, noise. QML algorithms are prone to the same problems (Weber et al., 2021; Liao et al., 2021; Guan et al., 2020; Lu et al., 2020; Ren et al., 2022).

Popular methods for adversarial defense (Madry et al., 2017) find it challenging to train large, robust classifiers, which are essential to solving real-world problems at the scale of ImageNet(Deng et al., 2009) or larger. These methods do not offer certifiable guarantees of robustness, even when they work well in practice.

Randomized smoothing is a state-of-the-art method that offers provable robustness against adversarial attacks without any assumptions about the underlying classifier. The defense works by aggregating a classifier's output in a region around the input - henceforth called the smoothing neighborhood - and computing the average probability of a class $\rho_c$. It is prohibitively expensive to compute the exact value of $\rho_c$ over the smoothing neighborhood since the number of points is exponential in the input dimension. In practice, Monte Carlo sampling algorithms are used to estimate $\rho_c$. Typically, randomized smoothing for adversarial robustness (Cohen et al., 2019; Yang et al., 2020; Lee et al., 2019) requires $N_{classical} \approx 10^5 - 10^6$ samples from the smoothing neighborhood.

**Contributions** In this paper, we discuss a purely QC approach to implementing randomized smoothing by using an orthogonal representation for the input space and use existing formalism for the Quantum Counting problem (Brassard et al., 2002). We create a superposition of the smoothing neighborhood of the input image and use our quantum circuit to output the average probability of prediction for a class $\rho_c$. We also design qubit state encoding and state preparation circuits for $l_0$, $l_1$ and other $l_p$ norm adversaries, and provide results from the simulation of the algorithm in Section 6.

**Theorem 1** *QuAdRo encodes an input $x$ into a quantum state $|\psi\rangle$ and, for error $\epsilon$ and confidence $1 - \delta$, requires total $M = O(1/\epsilon)$ queries to the base classifier $QNN_c$ to return certified radius for $x$. In comparison, any classical implementation of randomized smoothing based certification requires $M = O(1/\epsilon^2)$ queries for the same guarantees.*

Theorem 1 has been proved in Sec 4, and QuAdRo is presented in Alg 1.

## 2 RELATED WORK

### 2.1 RANDOMIZED SMOOTHING

Randomized smoothing (Cohen et al., 2019; Yang et al., 2020) method has achieved provable robustness against adversarial attacks. Given an input, one can define a smoothing neighborhood based on the threat model of the adversary described by $l_p$ norm and scale parameter $\lambda$. Such a robust model outputs the most likely class in the smoothing neighborhood returned by a base classifier, and this output is stable against $l_p$ perturbations. Cohen et al. (2019) first proved tight robustness guarantees for $l_2$ norm adversary using Gaussian smoothing. Later, Yang et al. (2020) provided guarantees for a larger set of adversaries and smoothing distributions, except $l_0$ norm, which was provided by Lee et al. (2019).

### 2.2 QUBIT STATE PREPARATION

Qubits are logical units of information for Quantum Computers, equivalent to bits in classical computers. Any QC device is made up of qubits that have the following two properties - superposition and entanglement. Superposition refers to a qubit's ability to exist in multiple states at the same time, while entanglement refers to the ability of multiple qubits to exist in a shared state such that an operation on one qubit also affects the state of another qubit instantaneously, without any additional transfer of information. Generally, $n$ qubits encompass a $2^n$ dimensional space where if bitstring $i = b_{n-1}...b_1 b_0$, then state $|i\rangle = \otimes_{j=0}^{n-1} |b_j\rangle$ where $b_j \in \{0, 1\}$.

There are numerous methods for encoding information into qubit states, often optimized for the target problem. For example, Novel Enhanced Quantum Representation (NEQR) (Zhang et al., 2013) and Flexible Representation of Quantum Images (FRQI) (Le et al., 2011) used for QC Image algorithms differ from variational heuristics for calculating molecular energies like Unitary Coupled-Cluster ansatz (Romero et al., 2018). Encoding methods popular in QML applications use the amplitude of a quantum state $|i\rangle$ as the representation of input vector element $x[i]$ to be encoded. This representation is really efficient but has the drawback that quantum search, amplitude estimation, etc., cannot be applied to such qubit states due to a lack of orthogonality. Amplitude encoding uses the same number of gates but a logarithmic number of qubits compared to the basis state encoding.

A number of distributions can be prepared as a superposed qubit state(Rattew & Koczor, 2022) - Log concave distributions (Grover & Rudolph, 2002), Uniform distribution using Quantum Fourier Transform (Deutsch, 1985), etc. Distribution parameters can be modified, either during state preparation, or using circuits like QADD (Koch et al., 2022) mid-circuit. QFT and inverse QFT, in particular, are a common pair of pre- and post-processing circuits that concentrate information from a superposition via a Fourier transform.

### 2.3 GROVER'S SEARCH ALGORITHM

Given a boolean objective function $f$ defined over unstructured space $S$ of size $N$ such that $\exists x \in S : f(x) = 1$ and a QC Oracle $O$ operator $O|x\rangle|y\rangle = |x\rangle|y \bigoplus f(x)\rangle$, Grover's Search algorithm (Grover, 1996) allows searching for $x$ in $O(\sqrt{N})$ calls to $O$. Using state preparation subroutine $U$

such that $U\ket{0} = \ket{\psi}$, the Grover diffusion operator

$$G = (2 * \ket{\psi}\bra{\psi} - \mathbb{1})O \tag{1}$$

is applied repeatedly to concentrate the set of desired outcomes. The result of the measurement of a concentrated state is a value $x \in S$ such that $f(x) = 1$ with an arbitrarily high probability. The original formulation requires $\ket{\psi}$ to be a uniform superposition of all values in the space, but later the same results were extended to arbitrary unitary operations $U$ (Biron et al., 1999). No QC unstructured search algorithm can perform better than $O(\sqrt{N})$ (Boyer et al., 1998).

### 2.4 QUANTUM AMPLITUDE ESTIMATION

An alternate formulation of Grover's search algorithm into a counting problem allows us to use the same number of oracle queries to estimate the number of solutions in $S$ for $f(x) = 1$ using Quantum Amplitude Estimation (QAE) (Brassard et al., 2002; Boyer et al., 1998). Variants of QAE (Grinko et al., 2021; Aaronson & Rall, 2020; Suzuki et al., 2020) tailored for low circuit depth and higher confidence use fewer oracle queries and/or fewer repetitions. The vanilla QAE in Figure 1 can be replaced with one of these variants without any loss of generality. A detailed comparison of these QAE variants can be found in Figure 3 in Grinko et al. (2021).

### 2.5 QUANTUM MACHINE LEARNING AND ADVERSARIAL ROBUSTNESS

There are multiple formulations of QML classifiers (Cong et al., 2019; Abohashima et al., 2020). Previous works have shown that QML models are prone to adversarial attacks. For example, (Guan et al., 2020) checks the robustness of QML models against noise in the training data using classical methods by modelling the verification problem as a classical SDP. In contrast, (Weber et al., 2021) implements new protocols for QML to certify robustness optimally. The threat models and algorithm in these papers differ from the algorithm presented here, though a QAE-based approach can provide a quadratic speedup in the case of (Weber et al., 2021) as well.

## 3 QUANTUM ADVERSARIAL ROBUSTNESS

As shown in Figure 1, the proposed algorithm for Quantum Adversarial Robustness (QuAdRo) uses state preparation $U^{p,\lambda}$ and Grover Diffusion Operator $G_c$ described in subsection 3.1 and 3.2. Using QAE, we can measure $\overline{\rho_c}$ - an estimate of $\rho_c$ - that base classifier answers correctly after smoothing, to high precision with high confidence, as discussed in subsection 4. The symbols in the paper are defined where they are first used, and a detailed table of notations is available in Appendix A.

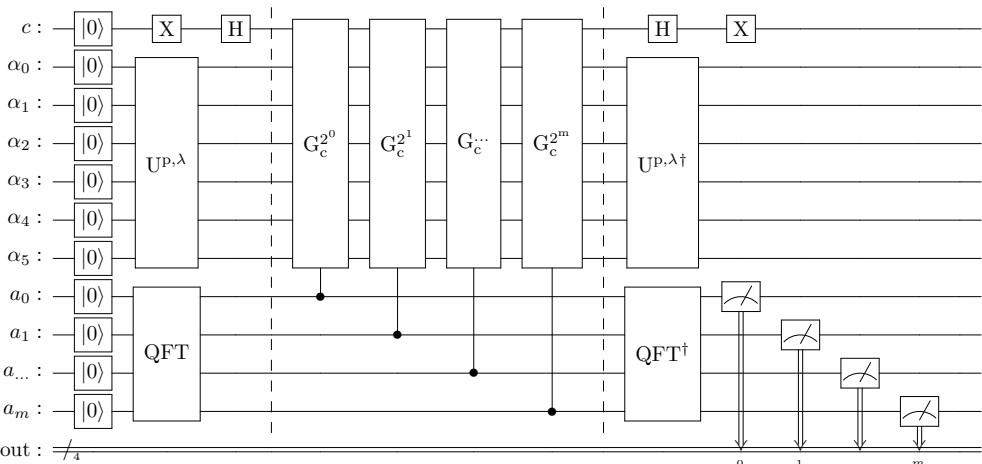

Figure 1: Quantum Estimation Circuit (QEC).

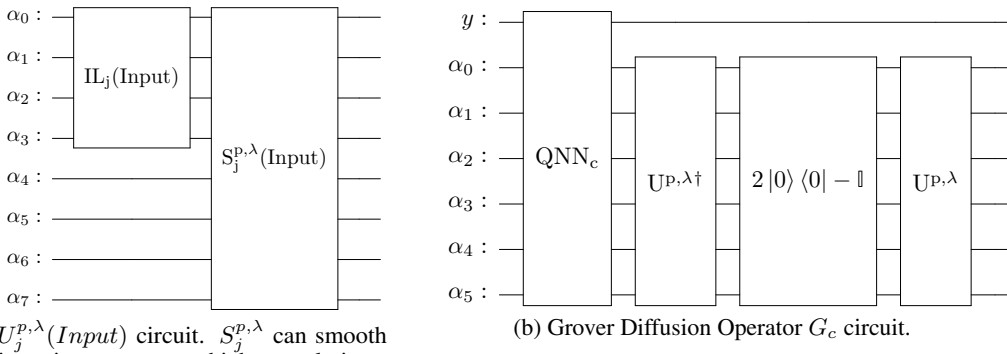

(a) $U_j^{p,\lambda}(Input)$ circuit. $S_j^{p,\lambda}$ can smooth the input image state to a higher resolution.

(b) Grover Diffusion Operator $G_c$ circuit.

Figure 2: Building blocks of QEC.

### 3.1 QUBIT STATE PREPARATION OPERATOR $U^{p,\lambda}$

Grover Diffusion Operator can only be used to search when distinct values in the input space are orthogonal. Here onwards, the smoothing neighborhood is represented as a superposition where the square of the amplitude of any perturbed input $|x + \varepsilon\rangle$ represents its probability in the smoothing distribution. Due to the requirements of orthogonality between individual inputs and representing the exponentially sized smoothing neighborhood, only an orthogonal basis state encoding can be used. This choice requires the same number of qubits as the number of classical bits.

Before state preparation, state qubits $\alpha$ and ancilla $a$ are reset to $|0\rangle$. First, the input loader circuit $IL$ loads the input $x$ from classical bits into qubit state $|\psi\rangle$. For the rest of the discussion, we define $d$ as the size of the input vector $x$, and $v$ is the number of qubits needed to represent any $x_j$. For a 28x28 grayscale image, $d = 784$ and $v$ is the resolution of a single pixel in the input, i.e. if each pixel belongs to range 0-255, i.e. $[0, 2^8 - 1]$, then $v = 8$.

Let $j^{th}$ value in input $x$ be $x_j = b_{j0}b_{j1}...b_{j(v-1)}$, i.e. $b_{ji} : i^{th}$ classical bit in $x_j$, then

$$|\psi\rangle = \otimes_{j=0}^{d-1} |x_j\rangle = |\alpha_0\alpha_1\alpha_2....\alpha_{dv-1}\rangle \tag{2}$$

To accomplish this, we use Pauli operators $\sigma_i$ ie. $\sigma_0 = \mathbb{1}$, $\sigma_1 = X$ to create input loader circuit $IL_j$ that loads $b_{ji}$ into qubit state $\alpha_{jv+i}$, defined as the tensor product of $\sigma_i$ as follows

$$IL_j = \otimes_{i=0}^{v-1} \sigma_{b_{ji}} \tag{3}$$

and by definition $IL_j |0\rangle^v = |\alpha_j\rangle$.

**Definition 1** *Input Loader operator $IL$ can be defined as*

$$IL = \otimes_{j=0}^{d-1} IL_j \tag{4}$$

*such that*

$$IL |0\rangle^{dv} = |\alpha_0\alpha_1\alpha_2....\alpha_{dv-1}\rangle = \otimes_{j=0}^{d-1} |x_j\rangle \tag{5}$$

Circuit $IL$ comprises of a maximum of $vd$ 1-qubit $X$ gates. For any input $x$, if $\|x\|_1 = k$, then $IL$ has exactly $k$ $X$ gates, corresponding to each $|1\rangle$ in $|\psi\rangle$.

After loading the input, a superposition of the smoothing neighborhood of the input $x$ is prepared based on the distribution $\phi(\cdot; \lambda)$ set to defend against a given $l_p$ norm adversary and parameterized by $\lambda$. Each perturbed value $x + \varepsilon$ is another valid input to the base classifier. For a detailed discussion about the design and functioning of the qubit encoding scheme, look at Appendix C.

**Definition 2** *Smoothing circuit $S^{p,\lambda}$ with distribution $\phi$ defined by $p$, $\lambda$, such that*

$$S_j^{p,\lambda} |j\rangle = \Sigma_{k=0}^{2^v - 1} \sqrt{\phi_j(\varepsilon_k; \lambda)} |x_j + \varepsilon_k\rangle \tag{6}$$

where $x_j + \varepsilon_k$ is a value in the neighborhood of $\alpha_j$. In our scheme, the complete smoothing operator $S^{p,\lambda}$ maps each value $x_j$ in the input vector independently to a probability-weighted superposition of its neighborhood. For any $i$ in the input space, probability weight of $i$ in the smoothed qubit state is $\phi(i; \lambda) = \Pi_j \phi_j(i_{jv} i_{jv+1} ... i_{(j+1)v-1}; \lambda)$. Hence

$$S^{p,\lambda} |\psi\rangle = \otimes_{j=1}^{d} S_j^{p,\lambda} |\alpha_j\rangle = \Sigma_{i=0}^{2^{dv}-1} \sqrt{\phi(i; \lambda)} |i\rangle \tag{7}$$

In this particular case, all $\phi_j$ are identical distributions centered around mean $x_j$. Detailed circuits for each $p$ norm are discussed in section 5

**Definition 3** *Qubit State Preparation Operator $U^{p,\lambda}$ (Figure 2a) comprises of $IL$ and $S^{p,\lambda}$*

$$U^{p,\lambda} = \otimes_{j=0}^{d-1} U_j^{p,\lambda} = \otimes_{j=0}^{d-1} S_j^{p,\lambda} IL_j \tag{8}$$

## 3.2 GROVER DIFFUSION OPERATOR $G_c$

Based on section 2.3 and 2.4, we design a robust QML classifier from any base binary classifier $f_c$ for class $c$ provided by a unitary quantum parallel (Nielsen & Chuang, 2002) oracle $QNN_c$. If output of the QML classifier for input $x$ is class $c$ then $f_c(x) = 1$ else $f_c(x) = 0$.

**Definition 4** *QML classifier oracle $QNN_c$*

$$QNN_c |x\rangle |y\rangle = |x\rangle \left| y \bigoplus f_c(x) \right\rangle \tag{9}$$

*Setting $|y\rangle = |-\rangle = \frac{1}{\sqrt{2}}(|0\rangle - |1\rangle)$ (Deutsch & Jozsa, 1992).*

$$QNN_c |x\rangle |-\rangle = (-1)^{f_c(x)} |x\rangle |-\rangle \tag{10}$$

When applying $QNN_c$ to the smoothed superposition of inputs given by $U^{p,\lambda} |0\rangle^{dv}$ from section 3.1, $\rho_c$ is the expected probability output by our algorithm i.e. input $|x + \varepsilon\rangle$ in the smoothing neighborhood belongs to class $c$

$$\rho_c = \mathbb{E}(\mathbb{P}(f_c(x + \varepsilon) = 1)) \tag{11}$$

**Definition 5** *The Grover Diffusion Operator $G_c$ derives from state preparation circuit $U^{p,\lambda}$ and $QNN_c$ as shown in Figure 2b.*

$$G_c = U^{p,\lambda}(2 |0\rangle \langle 0| - \mathbb{I})U^{p,\lambda\dagger} QNN_c \tag{12}$$

## 3.3 QUANTUM ESTIMATION CIRCUIT

QEC uses QAE (Brassard et al., 2002) to solve the counting problem defined for $QNN_c$ using $U^{p,\lambda}$ and $G_c$. The ancilla qubits $|a\rangle$ in Figure 1 are initialized in a uniform superposition in the range $[0, M]$ via QFT. Here, M is the total number of calls to the oracle in the circuit. Then, we apply controlled $G_c^{2^k}$ gates using $|a_k\rangle$ as control, which results in

$$QAE |\psi\rangle |a\rangle = G_c^a |\psi\rangle \otimes |a\rangle \tag{13}$$

for all $a$. After applying $QFT^\dagger$, we measure the ancilla state to obtain $\theta$ such that the probability measure $\overline{\rho_c}$ is

$$\overline{\rho_c} = sin^2(\theta/2) \tag{14}$$

## 4 THEORETICAL BOUNDS

As noted earlier, the smoothing distribution $\phi_j$ for each value in the input vector $x$ is independent, as per classical robustness criteria (Cohen et al., 2019; Yang et al., 2020). We claim that the proposed quantum algorithm QuAdRo finds $\rho_c$ with a quadratic speedup compared to the best-known classical algorithm with the same error bound. We show that the certified robustness problem can be reduced to a counting problem, and for a given error $\epsilon$ in the measurement of $\rho_c$ for fixed confidence $\delta$, the best known classical algorithm is $O(\frac{1}{\epsilon^2})$ and QuAdRo is $O(\frac{1}{\epsilon})$. This is also the best speedup possible for a counting problem using a quantum computer (Brassard et al., 2002).

---

**Algorithm 1** Quantum Adversarial Robustness (QuAdRo)

**procedure** QUADRO(Input $|\psi\rangle$, Class c, $U^{p,\lambda}, G_c, M, \delta$)
    $N_{rep} = 12\log\left(\frac{1}{\delta}\right) + 1$
    $N_{QEC} = \frac{M}{N_{rep}}$
    **for** i in Range($N_{rep}$) **do**
        $\theta_i = QEC1(|\psi\rangle, U^{p,\lambda}, G_c, N_{QEC}, \delta)$
        $\overline{\rho_c}\,[\text{i}] = sin^2(\frac{\theta_i}{2})$
    **end for**
    $\overline{\overline{\rho_c}}$ = median( $\overline{\rho_c}$ )
    lowerConfBound = $\overline{\overline{\rho_c}} - \frac{7}{N_{QEC}}$
    **if** $lowerConfBound \geq \frac{1}{2}$ **then return** c, CertifiedRadius(lowerConfBound)
    **else return** -1, ABSTAIN
    **end if**
**end procedure**

---

### 4.1 QUANTUM COMPUTING BOUNDS

Given that $m$ ancilla qubits are used for estimation, maximum number of oracle calls can be

$$M = \Sigma_{k=1}^m 2^k = 2^{m+1} - 1 \tag{15}$$

The probability that we measure $\theta$ correctly upto $m$ bits is $\frac{8}{\pi^2}$, and the measured probability value $\overline{\rho_c}$ is such that

$$\|\overline{\rho_c} - \rho_c\| \leq \epsilon_0 = 2\pi\frac{\sqrt{\rho_c(1-\rho_c)}}{M} + \frac{\pi^2}{M^2} \tag{16}$$

More generally, for a single experiment with no repetitions, the error in measurement of $\rho_c$ with a confidence $1 - \delta$ (using theorem 12 from Brassard et al. (2002)) is

$$\|\overline{\rho_c} - \rho_c\| \leq \epsilon = \epsilon_0 + \frac{1}{\delta}\left(\frac{\sqrt{\rho_c(1-\rho_c)}}{2M} + \frac{\pi^2}{4M^2}\right) + \frac{1}{\delta^2}\frac{\pi^2}{4M^2} \tag{17}$$

Solving Eq. 17 for M in terms of $\epsilon, \delta$ gives

$$M \approx \frac{\pi\sqrt{\rho_c(1-\rho_c)}}{\delta\epsilon} \sim O(\frac{1}{\delta\epsilon}) \tag{18}$$

The success probability $\frac{8}{\pi^2}$ in Eq. 16 can quickly be boosted to close to 100% by repeating the experiment multiple times and using the median estimate of $\rho_c$ (Miyamoto, 2022). As a result,

$$M \sim O(\frac{1}{\epsilon}log(\frac{1}{\delta})) \tag{19}$$

Constants for calculating M in Eq. 19 are small, as shown in Algorithm 11, and $N_{rep}$ and $N_{QEC}$ are based on Theorem 2 in Miyamoto (2022), also derived from Theorem 12 in Brassard et al. (2002).

### 4.2 CLASSICAL COMPUTING BOUNDS

There are numerous intervals and bounds commonly used for statistical guarantees, that result in a similar order of magnitude in calculating $M$ in terms of $\epsilon$ for a confidence $1 - \delta$. If we use Chernoff Bounds

$$\mathbb{P}(\|X - \mu\| \leq \delta\mu) \geq 1 - 2e^{-\frac{\mu\delta^2}{3}} \tag{20}$$

ie for $M$ repetitions of the function $f$ classically,

$$\mathbb{P}(\|\overline{\rho_c} - \rho_c\| \leq \epsilon) \geq 1 - 2e^{-\frac{M\epsilon^2}{3\rho_c}} \tag{21}$$

For the same setting, using Clopper-Pearson interval is a more popular practice since it provides tighter bounds upto $O(M^{-\frac{3}{2}})$ (Thulin, 2014).

$$\epsilon \leq M^{-\frac{1}{2}}z_{\frac{\delta}{2}}\sqrt{\rho(1-\rho)} + M^{-1} \tag{22}$$

For confidence $1 - \delta$, ignoring $\frac{1}{M}$ for large $M$,

$$\|\overline{\rho_c} - \rho_c\| \leq M^{-\frac{1}{2}} z_{\frac{\delta}{2}} \sqrt{\rho(1 - \rho)} \tag{23}$$

Hence

$$M \approx \frac{z_{\frac{\delta}{2}}^2 \rho(1 - \rho)}{\epsilon^2} \sim O(\frac{1}{\epsilon^2}) \tag{24}$$

## 5 CIRCUITS FOR SMOOTHING DISTRIBUTIONS $p, \lambda$

Table 3 shows best performing smoothing distributions for each common adversary (Yang et al., 2020; Lee et al., 2019). As shown in Yang et al. (2020), the same distribution can be used to counter multiple adversaries. These are a subset of all possible $(\phi, l_p)$ pairs, for which we present the state preparation circuits. Please refer to other works (Yang et al., 2020; Lee et al., 2019; Cohen et al., 2019) for detailed discussion on optimal distributions. Density in Table 3 should match the probability distribution obtained from state preparation $U^{p,\lambda}$. Appendix C discusses this topic in detail.

Table 1: Adversary norm parameters and corresponding certified robustness radius.

| Adversary $l_p$ | Distribution $\phi$ | Density | Certified Radius |
|---|---|---|---|
| $l_0$ | Tight Certificate | $x == \alpha \ ? \ \frac{\lambda}{2^v} : \frac{2^v - \lambda}{2^v(2^v - 1)}$ | $argmax_r \rho_r^{-1}(0.5) \geq p$ |
| $l_1$ | Uniform $l_\infty$ | $\propto \mathbb{I}(\|x\|_\infty \leq \lambda)$ | $2\lambda(\rho_c - \frac{1}{2})$ |
| $l_2$ | Normal Distribution | $\propto e^{-\|\frac{x}{\lambda}\|_2^2/2}$ | $\lambda GaussianCDF^{-1}(\rho; 0, 1)$ |

### 5.1 $l_0$ NORM

**Theorem 2** *The state preparation $U_j^{0,\lambda}$ in Figure 3 requires only $O(v)$ elementary gates ($\approx 2v$), where $cos(\frac{\theta}{2}) = \sqrt{\frac{\lambda - 1}{2^v - 1}}$ to prepare the state (proof in Appendix C).*

$$U_j^{0,\lambda} |x_j\rangle = \sqrt{\frac{\lambda}{2^v}} |x_j\rangle + \Sigma_{\beta \neq x_j} \sqrt{\frac{2^v - \lambda}{2^v(2^v - 1)}} |\beta\rangle \tag{25}$$

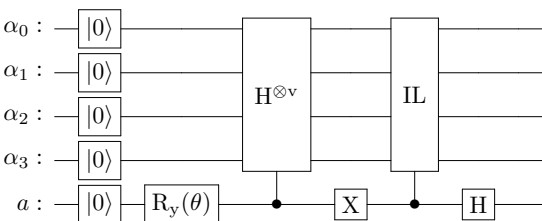

Figure 3: $U_j^{p,\lambda}$ circuit for $l_0$ norm adversary.

$U_j^{0,\lambda}$ smooths a state $|x_j\rangle$ into a quantum state with amplitude $\sqrt{\frac{\lambda}{2^v}}$ for $|x_j\rangle$ and an equal superposition of the rest of the space. $U_j^{0,\lambda}$ requires one additional ancilla qubit $a$ that cannot be disentangled.

### 5.2 $l_1$ NORM

For $l_1$ norm adversary, we use a uniform distribution with width $\lambda$, centred at the pixel value $\alpha$. To create a superposition state in a Uniform distribution, if $\lambda = 2^k, k \in \mathbb{Z}$, then the Hadamard operator $H$ on k-qubits $H^{\otimes k}$ can be used. The circuit in Figure 3 can also be adapted to prepare a uniform distribution by setting $\theta = \frac{\pi}{2}$. More generally, $QFT(\lambda)$ can prepare the requisite uniform distribution for any $\lambda$. The prepared uniform distribution can then be shifted to its corresponding mean, resulting in $U^{p,\lambda}$ as shown in Figure 4.

### 5.3 $l_2$ NORM

Creating Gaussian distribution (and any other log-normal distribution) is a well-studied problem (Grover & Rudolph, 2002). The relevant circuit is described in Figure 9 in the Appendix C. The same effect can also be achieved using the SHIFT($\alpha$, w) operator, as shown in Figure 4.

### 5.4 OTHER ADVERSARIES

Any log-concave distribution can be created as a qubit state with a quantum circuit (Grover & Rudolph, 2002), and the SHIFT operator in Figure 8 to shift the distribution to a mean value. Yang et al. (2020) uses log-concave distributions for randomized smoothing, for instance, Laplace or Uniform distribution for $l_\infty$ norm adversary. Refer to Table A.1 in Yang et al. (2020) for a comprehensive list of viable distributions and corresponding certified radii.

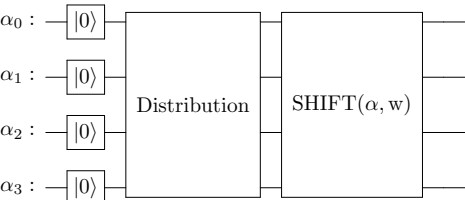

Figure 4: $U^{p,\lambda}$ circuit for $l_p$ norm.

## 6 EXPERIMENTS

Due to a lack of access to QC hardware at the time of writing, all the presented results are obtained using simulation. For the base classifier, we train a 2-layer Deep Neural Network (DNN) model for recognizing handwritten digits using the MNIST dataset from Torch v1.11.0. First, a fully connected layer is followed by leakyRelu activation, normalization, and scaling to 4-bit integer representation. This discretized classifier has an accuracy $\approx 89.8\%$ on the MNIST test dataset. We found that Projected Gradient Descent (PGD) adversary (Madry et al., 2017) can successfully attack this classifier, and this attack can be defended against using Randomized Smoothing (Cohen et al., 2019). Detailed model architecture can be found in Appendix B. All training and simulation used an Nvidia RTX 2080 Ti GPU.

To compare QuAdRo with the classical randomized smoothing algorithm, we reduce the problem size to simulate the QC circuit efficiently. First, we use the feature vector after layer 1 of the classifier as input that the adversary can attack. Second, we use a small layer 2 - $d = 5$. Third, we discretize feature vectors to 4-bit integer values after training - $v = 4$. This limits the input space to 20 bits, i.e. $2^{20}$ possible inputs, which can be completely simulated classically. Additional 6-12 ancilla qubits are required for the QEC. Based on preliminary experiments, the simulation cost of QEC-based prediction is high, so a classical predictor with $N_{predict} = 100$ is used in all experiments. Complete MNIST test set is used for experiments in Table 2, while simulation plots in Figure 5 use subsamples of size 1000 for 5a, b and 100 for 5c.

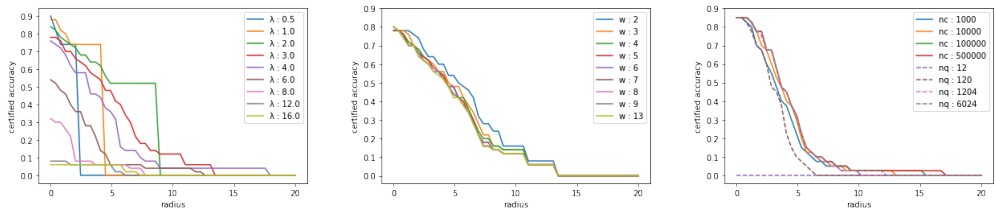

Figure 5: Certified accuracy vs radius for parameters - **a (left)** Adversary $\lambda$ for Normal distribution; **b (middle)** Smoothing distribution truncation parameter $w$ ; **c (right)** Oracle calls - nc : $N_{certify}$ for classical algorithm; nq : $N_{QEC}$ for QuAdRo;

The experiments are designed around four parameters - $\lambda$ adversary, $N_{QEC}$ calls to the classifier oracle, $1 - \delta$ confidence, and $2 * w + 1$ width of the truncated smoothing distribution. It is evident from Figure 5a that increasing $\lambda$ reduces certified accuracy as expected while a smaller $\lambda$ represents a weaker adversary. The effect of change in width $w$ on $\overline{\rho_c}$ in Figure 5b is not very pronounced. We observed that $w < \lambda$ cannot model a $\lambda$ adversary for radius $> w * d$. Since $w$ linearly increases simulation cost, $\lambda \in [2, 4]$, $w \in [4, 5]$ are optimal for our experiments. Table 2 shows the results of our experiments.

Table 2: Results : Accuracy of robust classifiers for MNIST test dataset - abstain (Abs) when $\overline{\rho_c} < 0.5$. Acc (!Abs) refers to the accuracy for certified inputs where classifier does not abstain.

| Classifier | Acc % | Abs % | Acc (!Abs) % | $\overline{\rho_c}$ (y = $y_{pred}$) | $\overline{\rho_c}$ (y $\neq$ $y_{pred}$) |
|---|---|---|---|---|---|
| Base | 89.73 | - | - | - | - |
| Quadro $N_{QEC}$ : 12 | 88.4 | 22.0 | 95.2 | 0.80 | 0.45 |
| Quadro $N_{QEC}$ : 120 | 88.4 | 17.7 | 94.1 | 0.83 | 0.39 |
| Classical $N_{certify}$ : 1000 | 87.7 | 17.8 | 94.0 | 0.82 | 0.40 |
| Classical $N_{certify}$ : 10000 | 88.0 | 17.8 | 94.0 | 0.83 | 0.40 |

Based on the theoretical discussion (Thulin, 2014; Montanaro, 2015), given the number of calls to the classifier oracle $M = N_{certify}$, we can use $N_{certify}$ samples for classical certification while QuAdRo uses $N_{rep}$ iterations of size $N_{QEC}$. Given $\epsilon_q \approx \frac{7}{N_{QEC}}$, $\epsilon_c \approx \sqrt{\frac{\rho_c(1-\rho_c)}{N_{certify}}}$, we should see a quantum advantage ($\epsilon_q < \epsilon_c$) for

$$N_{QEC} > 49 * \frac{12 \, log\frac{1}{\delta} + 1}{z_{\frac{\delta}{2}}^2 \rho_c(1 - \rho_c)} \tag{26}$$

QuAdRo's accuracy outpaces the classical algorithm as $N_{QEC}$), i.e. $N_{certify}$ increases, especially for contentious examples since they have a low certified radius. For median $\overline{\rho_c} = 0.8$ from Table 2, the results for the two algorithms match around $N_{QEC} \approx 2000$ for $\delta = 0.001$, which is observed in Figure 5c. To match the certified radius for $\rho_c \approx 1.0$, $N_{QEC}$ must scale by $\approx \frac{1}{1-\rho_c}$.

The simulations accurately depict the functioning of QuAdRo, except making $2^{dv}$ calls to the classifier to create the superposition state instead of $dv$ theoretically needed by the QC equivalent. In addition, floating point errors in the qubit state amplitude have been mitigated through normalization. All the theoretical bounds should remain the same for simulation results as well.

## 7   CONCLUSION

We developed the algorithm QuAdRo that creates a robust classifier and provides a certified radius for any base QML classifier. Section 4 establishes that the QuAdRo offers a quadratic speedup over existing classical algorithms for certified adversarial robustness via randomized smoothing. The same guarantees were simulated and tested in Section 6. The results hold for any distribution and classifier that can be implemented on a Quantum Computing device, and the QC circuit to prepare popular smoothing distributions as qubit states are presented in Section 5 and Appendix C. We have not made any additional assumptions on the base classifier; hence any guarantees that hold for prior art (Cohen et al., 2019; Yang et al., 2020) should hold for QuAdRo as well. Based on experiments, QuAdRo abstains less frequently and is more reliable than the robust classical classifier when $N_{certify}$ is increased.

**Broader Impact and Limitations:** The robustness guarantees in this paper apply only to the adversarial attacks on a classifier. The robust classifier will also inherit the underlying bias (e.g. training data, misrepresentation of objects/people, etc.) in the base classifier. The algorithm presented here also depends on a functional QC hardware for the speedup, and application to meaningful image inputs will need >1000 qubits.

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

## A    Notation and Schematic

Table 3: Notation used in the paper

| Symbol | Description |
|---|---|
| $x$ | Input (vector) |
| $b_{ji}$ | $i^{th}$ classical bit in $j^{th}$ element of the input vector |
| $\varepsilon$ | Perturbation to the input element |
| $c$ | Class predicted by base classifier |
| $l_p$ | Distance norm that characterizes the adversary |
| $\lambda$ | Scale parameter for smoothing distribution |
| $\phi\lambda$ | Smoothing distribution |
| $\rho_c$ | Average probability of class $c$ in the neighborhood of $x$ |
| $\overline{\rho_c}$ | Measured probability of class $c$ in the neighborhood of $x$ - using sampling / amplitude estimation |
| $\overline{\overline{\rho_c}}$ | Median of $\overline{\rho_c}$ |
| $M$ | Total number of calls to the base classifier (oracle) |
| $\epsilon$ | Error bound on $\overline{\rho_c}$ |
| $\delta$ | Confidence bound |
| $m$ | Number of ancilla qubits in QEC |
| $\lvert 0 \rangle$ | Ground state of qubit |
| $\lvert i \rangle$ | Bitstring for input $i$ encoded into qubits e.g $\lvert 5 \rangle = \lvert 0101 \rangle$ in 4 dimensional input space |
| $\alpha_k$ | $k^{th}$ qubit wire/state |
| $\sigma_i$ | Pauli operators ie. $\sigma_0 = \mathbb{I}$, $\sigma_1 = X$, $\sigma_2 = Y$, $\sigma_3 = Z$ |
| $\lvert \psi \rangle$ | Qubit state of Input |
| $IL$ | Image Loader QC operation to load $x$ into state $\lvert \psi \rangle$ |
| $S^{p,\lambda}$ | QC operation to prepare the superposition of smoothing neighborhood in $\lvert \psi \rangle$ |
| $U^{p,\lambda}$ | State preparation unitary based on $l_p$ and $\lambda$ |
| $QNN_c$ | QML classifier |
| $G_c$ | Grover diffusion operator for class $c$ |
| $QAE$ | Quantum Ampltitude Estimation |
| $QEC$ | Quantum Estimation Circuit |
| $QuAdRo$ | Quantum Adversarial Robustness algorithm |
| $N_{rep}$ | Number of repetitions of QEC in QuAdRo |
| $N_{QEC}$ | Number of classifier oracle calls in one shot/run of QEC |
| $\theta$ | Phase estimated by raw QAE operation s.t. $\overline{\rho_c} = sin^2(\theta/2)$ |

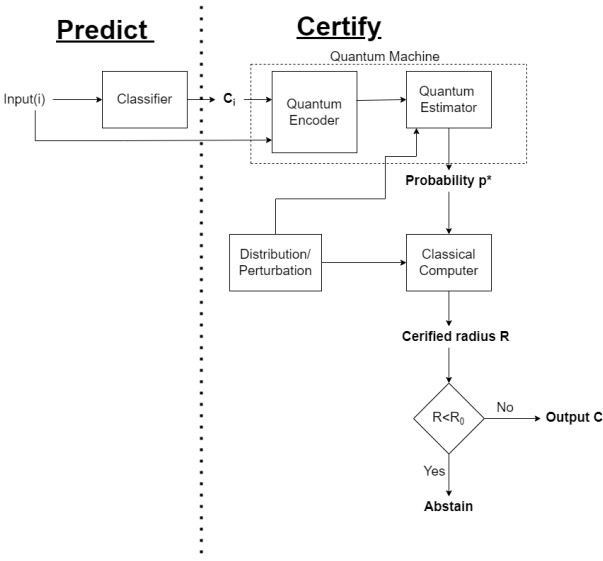

Figure 6: Flow diagram describing QuAdRo.

## B  SIMULATION : DISCUSSION, PLOTS, RESULTS

To be able to simulate QuAdRo on a classical computer, we reduced the problem size by following three changes - first, we used feature vector after layer 1 of the model as input that adversary can attack. Second, we use a small layer 2. Third, we discretize feature vectors to 4 bit int values after training. This allows us to limit the total input space to 20 bits, ie a space of $2^{20}$ possible inputs, which can be completely simulated.

Even for the smaller problem size, we show that

- The input space is local i.e. the nearby points in a neighborhood are likely to belong to the same class
- A non-trivial classical classifier $C$ can be trained on the dataset, achieving $\approx 90\%$ accuracy
- An adversary $A$ (PGD) can easily find inputs that are mis-classified by $C$
- Classifier based on randomized smoothing with $l_p$ norm and parameter $\lambda$ is effective against $A$'s attack.

We created the quantum algorithm simulator in numpy, using output of the $C$ over the complete input space of size $2^{20}$ to calculate the superposition of states and exact $\rho_c$ for comparison.

### B.1  MODEL ARCHITECTURE

First, we train a 2-layer DNN model - trained from scratch : 2 layer DNN, Layer 1 of size 784*5, Layer 2 of size 5*10, using LeakyRelu activation with a scaling factor 0.03. Layer 2 inputs are normalized, and scaled by $2^4$. Base classifier $C$ inherits trained Layer2 from model, and inputs are the feature vector output by Layer 1 in the previously trained model, discretized to 4 bit integer representation in range $[0, 16]$. Leaky Relu activation with a scaling factor 0.03 is used after fc1, to make feature vectors partially invertible and can be used to visualize images for features outside the train and test datasets.

### B.2  SIMULATION SETUP

We simulate QuAdRo for a 20 qubit input, with a total of 26-32 simulated qubits, since simulating a much larger system is not feasible on a classical computer. Due to lack of access to a quantum computer at the time of writing, the presented results were all simulated.

Given this setup, we implement both the classical and quantum computing algorithms for randomized smoothing to find the certified radius. We simulate Normal Distribution since it can defend against $l_1$, $l_2$, and $l_\infty$ adversaries (Yang et al., 2020). We show that for same number of calls to the classifier, QuAdRo error bounds improves and surpasses the classical algorithm for same hyperparameters and input.

The simulator implementation is non-trivial. We recommend reading the code (in supplementary material) and comments in addition to Appendix B. Following is a list of noteworthy features of the simulator

- Calculates MNIST classifier output for the complete input space. This is used to simulate $QNN_c$ oracle and calculate the exact value of $\rho_c$ and corresponding certified radius.
- Smoothing state $\psi$ is prepared using discretized Normal Distribution defined by

  ```
  f(i, mu, sd) = np.exp(-(i - mu)**2/(2*sd**2)
  ```

  for pixel value $mu$, $sd = \lambda$.
- The QFT operator on ancilla and the following Grover operator applied to $\psi$ have been simplified into an iterative operation on initial state. The iteration has $N_{QEC}$ steps.

### B.3  CODE SNIPPETS

We will release the code in supplementary material with its license. The code snippets here are only for reference, and we recommend using the code from supplementary material for simulation. The

following two functions can be used to simulate the Grover's Search operation by first preparing the requisite $\psi$, then repeating the search operation, or using it in a subroutine for amplitude estimation.

```
def get_psi_np(vals, low, high sd, v):
    p = []
    psi = 0
    for i, val in enumerate(vals):
        x = np.zeros(num_points)
        for i in range(low, high):
            x[i] = np.exp(-(i - mu)**2/(2*sd**2))
        p.append(x/np.linalg.norm(x))
        if(i == 0):
            psi = p[i]
        else:
            psi = np.kron(psi, p[i])
    norm = np.linalg.norm(psi)
    psi = psi/norm
    return (p, psi)

def grover(classifier_tensor, psi, prev_state):
    a = np.dot(psi, prev_state)
    cur_state = 2*a*psi - prev_state
    cur_state = classifier_tensor*cur_state
    # Normalize to deal with floating point issues
    norm = np.linalg.norm(cur_state)
    return cur_state/norm
```

### B.4 OTHER

One major difference in classical randomized smoothing and QuAdRo is the order of sampling and classifier operation. Classically, we sample from the neighborhood first, and then apply base classifier on each sample. In QuAdRo, we first apply the base classifier over the whole space, and then measure a posteriori. We expect to see a consistently better performance from QuAdRo for large $N_{QEC}$ than suggested by the theoretical discussions.

## C STATE PREPARATION CIRCUITS

### C.1 QUBIT ENCODING CHOICE

Classical data can be encoded into a quantum state in numerous ways. Amplitude encoding is one of the most popular methods - a vector of real/complex values is encoded as the amplitude of the $N$ states in a superposition such that the amplitude of $|i\rangle$ corresponds to the $i^{th}$ value in the input vector. Variations of this scheme have been used for different algorithms (Zhang et al., 2013; Le et al., 2011).

Oracle output for any input is pixel state dependent and any smoothing operation will destroy that information if pixel state was amplitude encoded - this is equivalent to running the algorithm for the mean of all images in the smoothing neighborhood for the given distribution.

### C.2 $l_0$ NORM

The circuit in Figure 7 is optimized for low circuit depth and gate count. General state of this system will be referred to as $|\psi_n\rangle = |x\rangle |a\rangle$, where $|x\rangle$ represents the qubits that will eventually be in a superposition state of the smoothing distribution, while $|a\rangle$ is the additional qubit needed in the circuit. Subscript $n$ represents $n^{th}$ logical step in the circuit.

Initially, all qubits are set to the ground state $|0\rangle$, hence the state of the system is

$$|\psi_0\rangle = |0\rangle^v |0\rangle \tag{27}$$

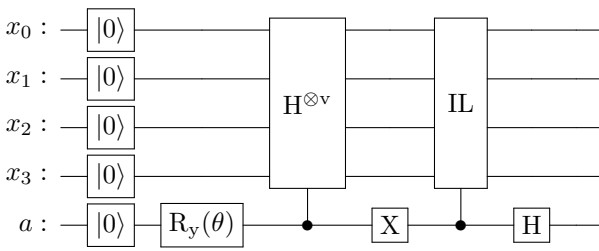

Figure 7: $U^{p,\lambda}$ circuit for $l_0$ norm adversary.

Assuming $V$ elements in the input space and adversary $\lambda$ such that $V > \lambda \geq 1$, the operator $R_y(\theta)$ rotates the ancilla state $a$, where $cos(\frac{\theta}{2}) = \sqrt{\frac{\lambda-1}{V-1}}$

$$R_y(\theta)|0\rangle = cos(\frac{\theta}{2})|0\rangle - sin(\frac{\theta}{2})|1\rangle \tag{28}$$

hence

$$|\psi_1\rangle = cos(\frac{\theta}{2})|0\rangle^v|0\rangle - sin(\frac{\theta}{2})|0\rangle^v|1\rangle \tag{29}$$

Applying $v$-dimensional Hadamard gate $H^{\otimes v}$ creates a uniform superposition of $V = 2^v$ states. Here, $H^{\otimes v}$ can be replaced with $QFT$ if $V \neq 2^v$ for generality. State after the $X$ gate is

$$|\psi_2\rangle = cos(\frac{\theta}{2})|0\rangle^v|1\rangle - \frac{1}{\sqrt{V}}sin(\frac{\theta}{2})(\Sigma_{i=0}^{V-1}|i\rangle)|0\rangle \tag{30}$$

After image loading,

$$|\psi_3\rangle = cos(\frac{\theta}{2})|\alpha\rangle|1\rangle - \frac{1}{\sqrt{V}}sin(\frac{\theta}{2})(\Sigma_{i=0}^{V-1}|i\rangle)|0\rangle \tag{31}$$

After Hadamard $H$ on the ancilla

$$|\psi_4\rangle = (cos(\frac{\theta}{2})|\alpha\rangle - \frac{1}{\sqrt{V}}sin(\frac{\theta}{2})\Sigma_{i=0}^{V-1}|i\rangle)|0\rangle - (cos(\frac{\theta}{2})|\alpha\rangle + \frac{1}{\sqrt{V}}sin(\frac{\theta}{2})\Sigma_{i=0}^{V-1}|i\rangle)|1\rangle \tag{32}$$

Applying QEC on $|\psi_4\rangle$ works for either value of a, with initial probability

$$\mathbb{P}(|x\rangle = \alpha) = cos^2(\frac{\theta}{2}) + \frac{sin^2(\frac{\theta}{2})}{V-1} = \frac{\lambda}{V} \tag{33}$$

## C.3 ALTERNATE FORMULATION

Alternatively, in a situation where repeating measurement of $|a\rangle$ is easier, we can use $cos(\frac{\theta}{2}) = \frac{\sqrt{V\lambda}-\sqrt{V+1-\lambda}}{V+1}$, and repeat the experiment until measurement $|a\rangle = 1$.

$$|x\rangle = (cos(\frac{\theta}{2}) + \frac{1}{\sqrt{V}}sin(\frac{\theta}{2}))|\alpha\rangle + \frac{1}{\sqrt{V}}sin(\frac{\theta}{2})\Sigma_{i\neq\alpha}|i\rangle \tag{34}$$

and probability of measuring ancilla

$$\mathbb{P}(|a\rangle = 1) = \frac{1}{2}(1 + 2(\frac{\theta}{2})sin\frac{\theta}{2}) \tag{35}$$

Hence

$$\mathbb{P}(a = 1) - \mathbb{P}(a = 0) = \frac{2}{(V+1)}(1 - \frac{(V-1)\sqrt{\lambda(V+1-\lambda)}}{V+1}) \tag{36}$$

Since $\mathbb{P}(a = 1) > \frac{1}{2}$, estimated number of repetitions of the state preparation circuit is less than 2. Please note that state preparation is also a part of Grover's Diffusion Operator, and will concentrate ancilla state $|1\rangle$ after repeated applications. In general, measuring the output of QEC circuit multiple (<2) times until $a = 1$ yields the required state. We expect this alternate circuit to be more useful as a component in a variational circuit.

To get the correct result, we need to perform the experiment multiple times, until ancilla state is measured as $|1\rangle$. The expected number of trials $<= 2$. In addition, ancilla with the discarded state $|0\rangle$ has an error $\sim O(\frac{1}{V})$ in the amplitude of $|x\rangle$ which may be acceptable for some $\lambda$, $V$, and $M$.

### C.4 SHIFT OPERATOR

Any prepared qubit state for a distribution can be shifted to a new mean using a quantum shift operator. The shift operator circuit $SHIFT\alpha$ based on QADD (Koch et al., 2022) shown in Figure 8 can take any superposition as input and shifts each $|i\rangle$ to $|i + \alpha\rangle$. We believe the same can be implemented using other quantum adders/circuits.

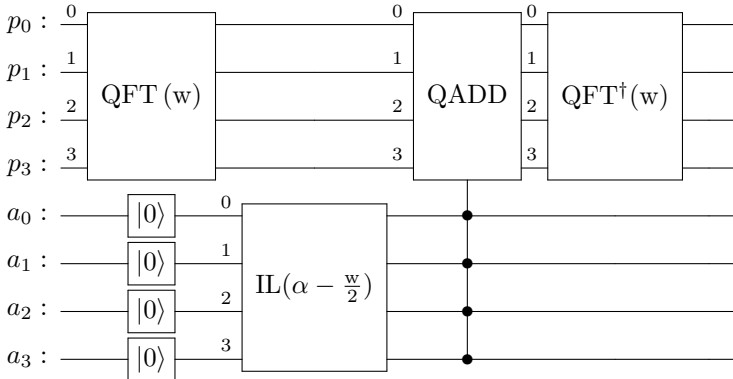

Figure 8: SHIFT($\alpha$, w) circuit to shift the input distribution $|p\rangle$ to mean $\alpha$.

### C.5 $l_2$ NORM

Basic Circuit for preparing $l_2$ norm distribution as a qubit state is shown in Figure 9.

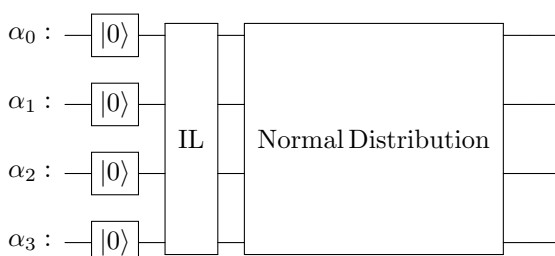

Figure 9: $U^{p,\lambda}(Input)$ implementation for $l_2$ norm (Grover & Rudolph, 2002).

The same effect can also be achieved using the SHIFT($\alpha$, w) operator, as shown in Figure 4.

## D QUANTUM MACHINE LEARNING CLASSIFIERS

There are multiple formulations that introduce variational quantum classification (VQC) circuits Cong et al. (2019); Farhi & Neven (2018); Havlíček et al. (2019); Schuld & Killoran (2019). A

parameterized quantum classification operation $V(;\gamma)$ on input $|\psi\rangle$ is defined as

$$V(;\gamma)|\psi\rangle = \Sigma_{ii}|\psi\rangle|0_i\rangle \tag{37}$$

Where $V(;\gamma)$ is a multilayer classifier which can be defined as

$$V(;\gamma) = V_1(;\gamma_1)V_2(;\gamma_2)...V_L(;\gamma_L) \tag{38}$$

The quantum classifiers in these works consist of a sequence of learnable, parameter-dependent unitary transformations $V_i(;\gamma_i)$ followed by a measurement operation of selected qubits that outputs the class. $V$ is trained so that $|_i|^2$ corresponds to the probability of input $|\psi\rangle$ belonging to class $i$.

Theoretically, the output of the system on input $|\psi\rangle$ is given by $\langle\psi|V^\dagger\mathbb{M}V|\psi\rangle$, where $\mathbb{M}$ is the measured observable at the end. For instance, measurement of output qubits in $\sigma_z$ basis can be used to estimate the predicted class. Experimentally, because of the probabilistic nature of the measurement, a high number of repetitions/shots need to be carried out in order to estimate this output distribution and predict the correct class.

Using a parameterized VQC as an oracle($_c$) within QuAdRo is non-trivial. Commonly used methods like Mid-Circuit Measurement and Reset (MCMR) or any other non-unitary operation before the end of QEC cannot be used. This restricts our choice to variationally trained classifiers that do not perform MCMR.

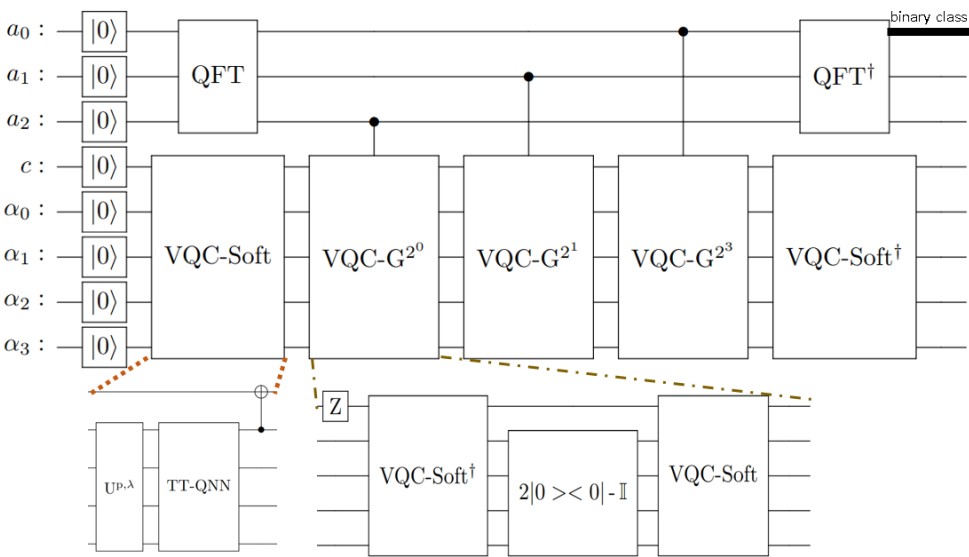

Figure 10: Hard classifier derived from TTQNN using nested QAE.

We implemented one such oracle by training TT-QNNZhang et al. (2020) using pennylane Bergholm et al. (2018) library. We trained both one vs all and multi-class classifiers for the reduced MNIST dataset. TT-QNN is a soft classifier by design where the output probability corresponding to class c in the output qubit state is mapped to extra ancilla qubits $a$ in a nested QAE operation within the oracle to convert it to a hard classifier oracle required for randomized smoothing, as shown in Figure 10. This construction increases the circuit depth and requires more oracle calls. On training, the multi-class classifier reached 74% accuracy for the same input features as used in Sec 6. We found that such a $QNN_c$ oracle is feasible, but simulation using existing libraries and available computational resources is prohibitively expensive. Hence, this oracle is not used in the presented results in this paper. We discuss this further in the supplementary material.

In another training approach, Salman et al. (2019) consider a generalization of randomized smoothing to soft classifiers $F$, which output a probability distribution over all classes for a given input, as compared to a hard classifier that outputs a single class. The adversarial training objective J in this case is given by(Equation 5 from Salman et al. (2019)),

$$J(x) = -log_{\varepsilon\sim\mathcal{N}(0,\lambda^2 I)}\mathbb{E}[F(x+\varepsilon)_y] \tag{39}$$

The training using adversarial examples of the smoothed soft classifer (SmoothAdv) improves the performance, which makes this another possible candidate for $QNN_c$.

Adversarial training of a VQC has been shown previously in Lu et al. (2020), where they extend classical methods of adversarial training like Projected Gradient Descent(PGD) to QML. To carry out adversarial training of a smoothed VQC, the training objective defined above can be optimised using the parameter shift rules as explained in Schuld et al. (2019).

