# OpenReview forum: "Adversarial Robustness based on Randomized Smoothing in Quantum Machine Learning "
_ICLR.cc/2023/Conference — Submitted to ICLR 2023_

### Official Review · Reviewer_gbVk · 2022-10-13

**Confidence:** 4
**Correctness:** 4
**Technical Novelty And Significance:** 2
**Empirical Novelty And Significance:** 2
**Recommendation:** 5

**Clarity, Quality, Novelty And Reproducibility:**

As I mentioned above, the clarity, quality, and novelty has space to improve both in terms of comparison to classical adversarial robustness literature, quantum Monte Carlo method, and quantum classification algorithm literature. In terms of reproducibility, the paper did a decent job by conducting detailed numerical experiments. In addition, the model architecture and simulation setup are explained in Appendix A.

**Strength And Weaknesses:**

From my perspective, the topic of studying quantum speedup of machine learning problems is of general interest. On the other hand, adversarial robustness also has wide applications. It’s nice to see the connection between these two topics and quantum can achieve quadratic speedup. In addition, the authors also conduct numerical experiments to demonstrate their result.

Nevertheless, I in general don’t find the contributions made by this paper competitive enough, both from the perspective of adversarial robustness and quantum computing. In current research on adversarial robustness, theoretical results have been obtained for many more scenarios, for instance deep learning (Salman et al. in NeurIPS 2019, https://proceedings.neurips.cc/paper/2019/file/3a24b25a7b092a252166a1641ae953e7-Paper.pdf), feature learning (Ilyas et al. in NeurIPS 2019, https://proceedings.neurips.cc/paper/2019/file/e2c420d928d4bf8ce0ff2ec19b371514-Paper.pdf), etc. This paper studies a primitive case with direct l_0, l_1, or l_2 norm adversarial noise, and the problem simply becomes a Monte Carlo problem of estimating a probability \rho_c = \sin^2(\theta/2) in Eq. (14). I think it would be very helpful if the authors could address why the results in this paper could be applicable to classical state-of-the-art results.

From the perspective of quantum computing, I have to say that the technical contribution is insignificant: it is a straightforward use of quantum amplitude amplification and estimation in Brassard et al. There have been many developments about quantum speedups of Monte Carlo methods, by Montanaro https://arxiv.org/abs/1504.06987, Hamoudi https://arxiv.org/pdf/2108.12172.pdf, etc. It would be nice to study whether more advanced quantum Monte Carlo methods could further improve adversarial robustness.

In addition, as adversarial robustness is in general studied in classification, the paper omits a main line of research on quantum classification algorithms. This is probably initiated by Kapoor, Svore, Wiebe, which studied quantum perceptron models for classification in NIPS 2016: https://proceedings.neurips.cc/paper/2016/hash/d47268e9db2e9aa3827bba3afb7ff94a-Abstract.html. Subsequently, Li, Chakrabarti, and Wu studied sublinear quantum algorithms with quadratic quantum speedup for linear and kernel-based classifications in ICML 2019: http://proceedings.mlr.press/v97/li19b.html, and Li, Wang, Chakrabarti, and Wu further studied  classification with quadratic quantum speedup with different norm schemes in AAAI 2021: https://ojs.aaai.org/index.php/AAAI/article/view/17028. Those results aim at general classification algorithms and do not cover adversarial, but at least a comparison to the current result in this paper and a potential discussion about the adversarial robustness in quantum classification algorithms will be very helpful. In terms of this, the last paragraph in the introduction claims that “Our novel contribution is showing how QC subroutines like Quantum Amplitude Estimation (QAE) can be used in ML” is an overclaim, because all these quantum classification algorithm papers have applied QAE.

Minor comments and suggestions:

Page 4: Between Eq. (7) and (8), there should be a . after “Detailed circuits for each p norm are discussed in section 5”. In addition, section 5 should be Section 5. When referencing theorems/lemmas/sections etc., the English word should be capitalized in general.

Page 5: In Eq. (14), should use math operators, sin -> \sin (otherwise it means the multiplication s*i*n). Similar corrections shall apply to psi, log, and many other math operators.

**Summary Of The Paper:**

Classification is one of the most applications in machine learning, and recently, an important research direction along this line is its adversarial robustness. Specifically, one of the state-of-the-art methods for defensing against adversarial attacks was achieved by randomized smoothing. This paper studies how to estimate the probability defined over the randomized smoothing neighborhood. Compared to classical algorithms with query complexity O(1/eps^2), the quantum algorithm has query complexity O(1/eps), which achieved a quadratic quantum speedup. Technically, this is achieved by quantum amplitude amplification and estimation.

**Summary Of The Review:**

In all, the problem of adversarial robustness by quantum algorithms is interesting in general, and this paper can be regarded as a first attempt for its quantum computing version. However, from my perspective, its contribution to the state-of-the-art understanding of machine learning is unclear and the technical contribution from the perspective of quantum computing is insignificant.

---

> ### Author Response · Authors · 2022-11-10
> **Response to Reviewer gbVk**
>
> Thank you for your recommendations.
>
> We have updated the paper based on the reviews - we have highlighted our contributions and provided main definitions and theoretical results as theorems. Earlier, we had defined each symbol where it was introduced. Now, we have added an additional table for all symbols in the Appendix for easy reference. We have also tried to clarify our claims, fixed typos, and simplified the notation.
>
> In an Appendix, we have also provided discussion on design and experiments with a QML oracle based on parametrized variational quantum circuits for classification introduced in previous works(https://arxiv.org/abs/1802.06002,https://www.nature.com/articles/s41586-019-0980-2, https://arxiv.org/abs/1803.07128). Among the other adversarial ML papers mentioned in the review, we show how to use soft-classifier adversarial training introduced in Salman et al (Neurips 2019, https://proceedings.neurips.cc/paper/2019/file/3a24b25a7b092a252166a1641ae953e7-Paper.pdf) since the the techniques are complementary to our approach. The other ML papers mentioned are more generic and do not have much overlap with adversarial robustness via randomized smoothing. Other quantum classification papers recommended in the review use quantum amplitude estimation as a subroutine, but they are unrelated to adversarial robustness as acknowledged by the reviewer themselves.
>
> Regarding other Monte Carlo methods beyond Amplitude Estimation, the other papers referenced in the review also default to the same approach as Brassard et al in the regime of randomized smoothing that we have discussed. We could not find any other relevant Monte Carlo techniques that add to our discussion. We have discussed that the encoding scheme as well as the Monte Carlo method used are optimal for an arbitrary classifier, and we cannot improve the performance without additional assumptions on the structure of the base classifier. Additional assumptions on classifier's structure would deviate from classical randomized smoothing.
>
> We have revised our claims and conclusion to be more clear and coherent. We believe our main contribution is our design choices and the end-to-end algorithm with the functional simulation code that can easily be verified for a real-world ML problem.
>
> Though the reviewer's analysis is fairly accurate, we believe that our contribution in the data loading and smoothing procedures that connect randomized smoothing-based adversarial robustness to Quantum Amplitude estimation is non-trivial even if it appears evident after the fact. For instance, identifying that amplitude encoding of data cannot be used due to the need to ensure orthogonality of distinct inputs is crucial. Identifying that the noise used for randomized smoothing lacks correlation between two pixels makes the presented work possible. We also develop a new circuit design for loading the noise distribution that offers robustness against $l_0$ norm adversaries. Our input encoding choices make it possible to adopt existing techniques in loading the noise distributions centered around arbitrary mean values, which is essential for the procedure to work.
>
> We thank reviewer for their detailed comments and hope to further discuss the avenues for improvement. Kindly let us know if you have other suggestions or anything we didn't address.

---

> > ### Comment · Reviewer_gbVk · 2022-11-26
> > **Official reply**
> >
> > Thanks for the discussions. I think the current version indeed improved, and I slightly increased my score. Nevertheless, given my concerns as well as those mentioned by other reviewers, this paper is still probably not good enough for the high bar of ICLR.

---

### Official Review · Reviewer_BnBR · 2022-10-15

**Confidence:** 3
**Correctness:** 3
**Technical Novelty And Significance:** 2
**Empirical Novelty And Significance:** 2
**Recommendation:** 6

**Clarity, Quality, Novelty And Reproducibility:**

Clarity: This work goes through the proof smoothly from preliminaries to the point it focuses.

Quality: Since this work mainly works on derivation of bounds, it looks good to me.

Novelty: This work is an incremental work on quantum amplitude estimation and quantum adversarial robustness, so I would say it is less novel while it provides experimental comparison.

**Strength And Weaknesses:**

Strength:

1. This work well introduces preliminaries randomized smoothing and related quantum computing concepts.

2. This work derivates operators of quantum adversarial robustness and its theoretical bounds carefully. It is well explained to make reader from adversarial ML with limited knowledge on quantum computing easier to capture the high-level ideas.

3. This work considers different considers lp norm adversaries.

Weakness:

1. This work is built upon existing quantum computing algorithms to address the adversarial robustness issue in QML, which may look less innovative.

2. Dataset used in this work is quite out-dated in the ML community and NN with 2 layers is also shallow in the practice. If the theoretical work would guide more applications, the dataset and NN should be more complex.

3. Since the experiments of quantum part are reduced to simulation of QC circuits, it would be nice to provide bounds for the simulation as well.

**Summary Of The Paper:**

Adversarial attack is a common issue in traditional ML models against malicious adversary. Randomized smoothing is a sota method to tackle the issue. However, computing exact probability over the smoothing neighborhood is computationally expensive. Instead, sampling is required to estimate the probability. This work focuses on its quantum computing counterpart by applying quantum amplitude estimation with a quadratic speedup compared to existing quantum adversarial robustness. It designs qubit state encoding from classical input to the qubit states, and state preparation circuits for smoothing distributions.

**Summary Of The Review:**

This work focuses on adversarial robustness of quantum ML. It mainly develops the area from theoretical perspective. While it is more like an incremental work based on some existing algorithms, we should encourage to have more theoretical works like this in ML community. It is good to come up with solid computational bounds first, which will provide capability of related algorithms and guide more applications later on.

---

> ### Author Response · Authors · 2022-11-14
> **Response to Reviewer BnBR**
>
> Thank you for your recommendations.
>
> 1. The contributions may look less innovative, especially since we have tried to reference the slightest inspiration to its source. That said, we have extensively explored various design choices, provided theoretical bounds, and a detailed algorithm for Quantum Adversarial Robustness.
>
> 2. Our goal with the experiments was to demonstrate the validity of QuAdRo for a functional base classifier. The reviewer is correct in pointing out the limitations of the minimal 2-layer NN and outdated dataset, yet the NN possesses all the properties that make it a good example -
>
>     a.) Small enough feature vector space (20 qubits) that can be simulated without expensive hardware
>
>     b.) Easily available adversarial examples in the neighborhood
>
>     c.) Reasonably high accuracy ~90%
>
>     d.) Reproducibility and lack of complicating factors that allow readers to see both the strengths and weaknesses of our contribution
>
> 3. We have updated the paper for clarity and explained the limitations introduced by simulation vs actual QC hardware. To simulate the superposition quantum state based on the smoothing neighborhood, we need an exponential number of queries to obtain classifier output for each point in the neighborhood. However, the output of each simulation step is theoretically the same as it would be on a QC device, barring factors beyond our control, like qubit fidelities, limits of device precision etc. We will provide more information if the reviewer can elaborate on what bounds they expect.

---

### Official Review · Reviewer_HtzH · 2022-10-24

**Confidence:** 3
**Correctness:** 3
**Technical Novelty And Significance:** 3
**Empirical Novelty And Significance:** 3
**Recommendation:** 6

**Clarity, Quality, Novelty And Reproducibility:**

Clarity score 4/10: This paper is in general not clearly written. See the Weakness for example.
I also have the following suggestions:

C1.1) It is also suggested to give a list of notations before introducing how to design the quantum algorithm.

C1.2) Another suggestion is to tell the reader the dimensionality of a vector or matrix in Eqs. (2)-(13).

Quality score 5/10: The derivation of query complexity seems correct. It is suggested to add references to support Eq. (16).

Novelty score 5/10: The proposed quantum randomized smoothing is moderately new, because techniques like randomized smoothing in classical ML, Grover-like operators, quantum preparation of distributions, etc. have already been well developed.

Reproductivity 8/10: I run the shared code and it works.

**Strength And Weaknesses:**

Strength:

S1) By using randomized smoothing, a new method is proposed for adversarial robust quantum machine learning.

S2) The proposed quantum randomized smoothing need $O(1/\epsilon)$ queries of the classier compared to the complexity  $O(1/\epsilon^2)$ of its classical counterpart.

S3)  Experimental results showed the effectiveness as well as efficiency of proposed method.

Weakness:

W1) The proposed method cannot be applied to real applications due to the bottlenecks in the development physical quantum computers. As discussed in the conclusion, the algorithm presented here is also dependent on a functional QC hardware for the speedup, and application to meaningful image inputs will need >1000 qubits.

W2) This paper has some parts not well written.

There are some examples:

(W1.2.1) In Figure 1, there are $m$ Grover operators but the authors use 4 ancilla quantum bits $a_0,\cdots,a_3$. It is suggested to make them consistent. Also, the operator $G_{c}{\cdots}$ should be in its correct form.

(W1.2.2) In Eq. (7), the (Kronecker) basis $|i\rangle$ is suggested to be explained for ML audiences who are not familiar with quantum symbols.

(W1.2.3) In Eq. (10), it is suggested to give the explicit form of $|-\rangle=\frac{1}{\sqrt{2}}(|0\rangle-|1\rangle)$.

(W1.2.4) In Eq. (13), "psi"-->$\psi$.

(W1.2.5) Why choose parameter $\frac{7}{N_{QEC}}$ in Algorithm 1?

(W1.2.6) In P12, "space of size $2^20$"--> "space of size $2^{20}$", "exact value of $rho_c$"-->"exact value of $\rho_c$".

**Summary Of The Paper:**

This paper proposes a randomized smoothing method for adversarial robust quantum machine learning. The authors uses superposition of "random noises" followed by Grover-based counting in designing quantum circuits. The proposed quantum randomized smoothing need $O(1/\epsilon)$ queries of the classier compared to the complexity  $O(1/\epsilon^2)$ of its classical counterpart. Experimental results showed the effectiveness.

**Summary Of The Review:**

Due to the comments in "Clarity, Quality, Novelty And Reproducibility", I suggest "marginally above the acceptance threshold".

-------------After rebuttal----------------
I am so sorry for my carelessness of incorrectly choosing "5 borderline reject" as the suggested score.
Many thanks to the authors for their feedback. The revision clearly addressed some of my concerns, and  I would like to choose "6 borderline accept".

---

> ### Author Response · Authors · 2022-11-10
> **Response to Reviewer HtzH**
>
> Thank you for your recommendations and for pointing out our errors and typos.
>
> W1.2.5 is based on theorem 2 from Miyamoto et al (https://arxiv.org/abs/2108.09014). We have highlighted this in the discussion.
>
> We have updated the paper to improve the clarity of presentation and removed inconsistencies as suggested. We have provided a table of notation in the Appendix for quick reference and added additional remarks to highlight our design choices and the important components of the circuit. Please let us know if you have other suggestions.
>
> We noticed that you suggested "*marginally above the acceptance threshold*" in the summary, but the recommendation score (5:marginally below the acceptance threshold) doesn't match the summary. Please update the review if this is unintentional.

---

> ### Author Response · Authors · 2022-12-07
> **Response to Reviewer HtzH**
>
> _1. Summary Of The Review: Due to the comments in "Clarity, Quality, Novelty And Reproducibility", I suggest "marginally above the acceptance threshold"._
>
> We noticed that you suggested "marginally above the acceptance threshold" in the summary, but the recommendation score (5:marginally below the acceptance threshold) doesn't match the summary. Please update the score if this is unintentional.

---

### Official Review · Reviewer_fJTC · 2022-10-25

**Confidence:** 2
**Correctness:** 3
**Technical Novelty And Significance:** 2
**Empirical Novelty And Significance:** Not applicable
**Recommendation:** 5

**Clarity, Quality, Novelty And Reproducibility:**

I think the paper would benefit a lot from careful proofreading to fix typos, language, and notation issues.  I am not an expert in Quantum Computing and Quantum ML however, I think that the results presented in this work are not particularly novel and original.

**Strength And Weaknesses:**

Strengths

1. This work attempts to show how quantum computing primitives can be used to improve standard machine learning tasks such as adversarial training.


Weaknesses

1. I do not think that the paper is well-written. The majority of the ICLR community (myself included) are not experts in quantum computing and therefore a more detailed presentation of the results and contributions as well as the techniques would greatly help this paper. Apart from the presentation, there are various typos and language issues and notation is not properly defined (e.g., the notation in most equations of the main body).

2. It seems that the main contributions of this work are theoretical and yet the main results have no formal statements and seem to follow directly from standard QC results (see the results in Section 4.1).



**Summary Of The Paper:**

This work considers the problem of designing quantum algorithms to perform adversarial training of classifiers using randomized smoothing.  Using classical method the number of required samples (and therefore queries to the classifier) is roughly $O(1/\epsilon^2)$, where $\epsilon$ is the target error parameter for the expectation over the smoothing distribution.  In this work, the authors use quantum computing primitives to design a randomized smoothing algorithm that only requires $O(1/\epsilon)$ samples, improving over the classical.

**Summary Of The Review:**

This work considers the problem of designing quantum algorithms to perform adversarial training of classifiers using randomized smoothing.  In this work, the authors use quantum computing primitives to design a randomized smoothing algorithm that has improved sample complexity over the corresponding classical approach. Applying QC methods in ML is an interesting direction that has a lot of potential but, in my opinion, this paper in its current state is not ready for publication.  The authors should revise the manuscript and more carefully define notation.  They should also try to present more formally their theoretical results so that their contributions are clear.  I am currently inclined toward rejection but since I am not an expert in this area I am open to discussion with the other reviewers and the authors.

---

> ### Author Response · Authors · 2022-11-14
> **Response to Reviewer fJTC**
>
> Thank you for your recommendations.
>
> We have updated the paper based on the reviews - we have highlighted our contributions and provided main definitions and theoretical results as theorems. Earlier, we had defined each symbol where it was introduced. Now, we have added an additional table for all symbols in the Appendix for easy reference. We have also tried to clarify our claims, fixed typos, and simplified the notation.
>
> The main contribution of this work is the algorithm and the individual components that make the existing theoretical results (Grover's) apply to the problem of Certified Adversarial Robustness via Randomized Smoothing. We provide the theoretical analysis and experimental results for the same. We have updated relevant parts in the paper to make our contributions more clear.
>
> We welcome further comments and discussion.

---

### Author Response · Authors · 2022-11-17
**Request for discussion**

Based on all reviews, we have made the requisite changes in the paper and provided detailed responses to address the other points raised by each reviewer. In light of the approaching deadline, we request all the reviewers to start a discussion as soon as they can. We look forward to your comments on our updated submission.

---

### Decision · Program_Chairs · 2023-01-20

**Decision:**

Reject

**Justification For Why Not Higher Score:**

see summary above

**Justification For Why Not Lower Score:**

N/A

**Metareview: Summary, Strengths And Weaknesses:**

The reviewers generally felt the paper was a bit below the bar for ICLR in terms of novelty, both from a quantum and an adversarial robustness perspective. Some of the theory results were thought to be incremental, and there were also concerns about experiments and practicality, as well as presentation. The author response helped a little bit, but the concerns were not completely removed.